# Regulatory Standard for Determining Preoperative Skin Preparation Efficacy Underreports True Dermal Bioburden in a Porcine Model

**DOI:** 10.3390/microorganisms12112369

**Published:** 2024-11-20

**Authors:** Hannah R. Duffy, Nicholas N. Ashton, Abbey Blair, Nathanael Hooper, Porter Stulce, Dustin L. Williams

**Affiliations:** 1Department of Orthopaedics, University of Utah, Salt Lake City, UT 84112, USA; hannah.duffy@utah.edu (H.R.D.); n.ashton@utah.edu (N.N.A.); abbey.blair@hci.utah.edu (A.B.); hooper.33@wright.edu (N.H.); u1270386@utah.edu (P.S.); 2Department of Biomedical Engineering, University of Utah, Salt Lake City, UT 84112, USA; 3Department of Pathology, University of Utah, Salt Lake City, UT 84112, USA; 4Department of Physical Medicine and Rehabilitation, Uniformed Services University of the Health Sciences, Bethesda, MD 20814, USA

**Keywords:** preoperative skin preparation, surgical site infection, skin microbiome, cup scrub method, tissue blend method, porcine model

## Abstract

Medical device companies and regulatory bodies rely on a nondestructive bacterial sampling technique specified by the American Society for Testing and Materials (ASTM E1173-15) to test preoperative skin preparations (PSPs). Despite the widespread use of PSPs, opportunistic skin-flora pathogens remain the most significant contributor to surgical site infections, suggesting that the ASTM testing standard may be underreporting true dermal bioburden. We hypothesized that ASTM E1173-15 may fail to capture deep skin-dwelling flora. To test this hypothesis, we applied ASTM E1173-15 and a full-thickness skin sampling technique, which we established previously through application to the backs of seven pigs (Yorkshire/Landrace hybrid) following a clinically used PSP (4% chlorhexidine gluconate). The results showed that samples quantified using the full-thickness skin method consistently cultured more bacteria than the ASTM standard, which principally targeted surface-dwelling bacteria. Following PSP, the ASTM standard yielded 1.05 ± 0.24 log_10_ CFU/cm^2^, while the full-thickness tissue method resulted in 3.24 ± 0.24 log_10_ CFU/cm^2^, more than a 2 log_10_ difference (*p* < 0.001). Immunofluorescence images corroborated the data, showing that *Staphylococcus epidermidis* was present in deep skin regions with or without PSP treatment. Outcomes suggested that a full-thickness sampling technique may better evaluate PSP technologies as it resolves bioburdens dwelling in deeper skin regions.

## 1. Introduction

Despite the clinical use of preoperative skin preparations (PSPs) through topical antisepsis, the patient’s endogenous flora is the primary source of surgical site contamination [1,2,3,4]. Culture data from surgical instruments, surgeons’ gloves, patients’ tissues, and hardware point to the patient’s microflora as the principal source of microbial growth [5,6,7,8]. PSP-surviving microbes account for 70–95% of surgical site infections (SSIs), greatly outnumbering contributions from exogenous sources [1]. As such, SSI isolates mirror the natural geography of the human skin microbiome. For example, surgeries involving the shoulders, breasts, and spine incur much higher rates of infection with *Cutibacterium acnes*, a common resident of these locations [9,10,11,12]. In all other anatomical regions, Gram-positive staphylococcal species such as *Staphylococcus epidermidis* and *Staphylococcus aureus* are principal SSI contributors [13,14,15]. If this persistent surgical site contamination is to be addressed, the nature of PSP-surviving microorganisms and how they evade prophylactic antiseptics must inform the development of future testing methods, surgical practices, and next-generation antiseptic technologies.

Thousands to millions of bacteria per cm^2^ live throughout the skin’s surface and deeper layers [16,17,18,19]. Both superficial and deep-dwelling bacteria have been identified using various imaging techniques, including Gram stain and fluorescent imaging [20,21,22]. As early as 1985, Gram stains showed the presence of bacteria in hair follicle tracts and at the edge of the pilosebaceous glands [22]. In recent years, researchers have used peptide nucleic acid fluorescence to show bacterial aggregates on the stratum corneum and hair follicle bulbs in healthy controls [20]. This investigation also quantified the bacterial skin aggregates in biofilm: 92% of healthy patients had observable biofilm, with the majority residing in the hair follicles (64%) or the stratum corneum (36%) [20]. Analyses of the relative abundance of the 16S rRNA gene (common to all bacteria) also indicated that the depth profiles of bacteria vary on skin characteristics and the presence of skin features [23]. On palm skin (hairless), bacteria are more plentiful in the first 0–1 mm of the skin than depths greater than 1 mm [23]. In contrast, human facial skin (with hair) exhibits more bacteria below 3 mm than between 1 and 2 mm [23]. Human hair follicles, which harbor resident normal flora, extend 2–4.5 mm below the skin surface [24]. Likewise, sebaceous glands and sweat glands extend into the dermis and provide environments for bacteria growth [22,23]. Bacteria living in skin features across all dermal layers make up the human skin microbiota and contribute to SSIs, a fact that researchers have been exploring for nearly 70 years [25,26,27].

The use of PSPs have dramatically reduced microbial bioburden and thus SSI rates to 1–4% since their introduction as part of broader antiseptic techniques [28]. Current skin disinfection with PSP is performed a few minutes before surgery in combination with other SSI-preventing practices [29,30,31]. PSP products are effective on the skin’s surface, often resulting in log_10_ bacterial reductions in human subjects [32,33]. PSP products with chlorhexidine gluconate (CHG) have become the clinical gold standard due to CHG’s long-acting efficacy [34]. CHG and other antiseptics are highly potent upon contact with bacteria. However, rapid antisepsis before surgery does not kill bacteria in the deeper regions of the skin due to the depths of bacterial colonization [26,35]. It is these deeper-dwelling microorganisms that are often overlooked by PSP testing practices.

Determining true PSP efficacy using appropriate testing methodologies to address deeper-dwelling microbes is essential as bacteria that survive PSP pose a significant SSI risk. The industry relies on the American Society for Testing and Materials (ASTM) standard E1173-15, Standard Test Method for Evaluation of Preoperative, Precatheterization, or Preinjection Skin Preparations, when testing PSP products [36]. Generally regarded as the cup scrub method, this method has been the gold standard for PSP development since 1994 when 59 Federal Register 31402 was created [37]. Briefly, the cup scrub method involves placing a sterile cup on the skin’s surface [37,38]. The cup is filled with solution. A rubber spatula is used to rub the skin’s surface for approximately 1 min. Then, the liquid is removed from the cup, serially diluted, and quantified to detect microorganisms suspended in the liquid. This standard specifies levels of acceptable bioburden reduction as 2 and 3 log_10_ units for dry and moist sites, respectively. PSP products used clinically are often approved by the Food and Drug Administration (FDA) based on cup scrub method outcomes.

The cup scrub technique fits into a broader category of nondestructive methods including swabs, scraping, and impressions. Generally, the cup scrub method is one of the more thorough nondestructive methods for quantifying skin flora, as repeated scraping may dislodge more bacteria than a swab. Of all nondestructive methods, the cup scrub method has become especially consequential through sanctioning by the FDA. The widespread use of the cup scrub method in antisepsis, skin conditions, and cosmetics demonstrates the need for skin sampling methods [39,40,41,42,43,44]. Despite its prevalence, the cup scrub method fails to consider bacteria living below the skin’s surface.

Unlike nondestructive techniques, destructive or biopsy-like methods better capture microorganisms throughout various skin layers [22]. These methods are rarely performed, however, as biopsies or other destructive techniques result in skin trauma for patients [45]. Moreover, studies that do use biopsies may produce confounding results conditional on the anatomical location and size of the biopsy. Depending on the quantity of surface area used, variation across the skin’s surface can yield a high bioburden in some samples, while others may yield no growth [46]. Thus, manufacturers of antiseptic products opt to demonstrate substantial equivalence through the 510(k) FDA pathway [47]. Generations of 510(k) predicates have perpetuated the use of the cup scrub method that fails to resolve microbes of all skin layers. Destructive techniques are the only techniques that have demonstrated sensitivity to deeper-dwelling microbes. Iterative destructive testing for PSP products at scale is possible using a large animal model.

We previously established the tissue blend method using a pig model, wherein we removed and processed 4 × 4 cm (16 cm^2^) full-thickness sections of dermal tissue (surface to fascia) for testing PSPs [48]. This method addressed the major disadvantages of nondestructive methods. By homogenizing full-thickness tissue, we resolved bacteria living in all skin layers and maximized the number of detectable microbes using optimal surface area and depth profiles. The 4 × 4 cm excisions included a surface area roughly 14× larger than a standard large biopsy punch. Surface area is a critical consideration of a PSP model as bacteria are dispersed in a nonhomogeneous pattern throughout the skin. Bacteria are found in pockets throughout the skin’s varied anatomical features, including hair follicles, sweat glands, and sebaceous glands, but are not organized systematically [1,16,22]. Samples with larger surface areas incorporate more features housing bacteria, resulting in a more accurate average. We employed this model to test two gold-standard PSPs with povidone-iodine and CHG [48]. Neither antiseptic achieved a 2 log_10_ reduction, highlighting the need for improved PSP technology. The tissue blend method can be readily used to test current and new PSP approaches in an animal model by using full-thickness tissue and large surface areas.

In this study, we hypothesized that the ASTM standard E1173-15 would overlook deep-dwelling bacterial communities compared to a previously established full-thickness skin sampling method. If our hypothesis is supported, the tissue blend method is a good model to test improved PSP technology with the long-term goal of targeting the problem that underpins SSIs.

## 2. Materials and Methods

### 2.1. Supplies, Reagents, and Instruments

Alcohol-based Wet Skin Scrub Trays with CHG (CHG preoperative skin preparation kit, 4% solution) were purchased from Medline (Northfield, IL, USA). Stainless steel cups were custom fabricated from corrosion-resistant tubing purchased from McMaster-Carr (Elmhurst, IL, USA) using 316 stainless steel (with a 1/4” wall thickness and a 1–1/2” outer diameter, 89495K49). The cups were cut to approximately 1 in. in height, sanded, and polished. Glass rods with rubber policemen scrapers (spatulas) were purchased from Cole-Parmer (Vernon Hills, IL, USA). Columbia blood agar plates were acquired from Hardy Diagnostics (Santa Maria, CA, USA). Petri dishes and agar were purchased from Fisher Scientific (Hampton, NH, USA). Dey-Engley (D/E) neutralizing broth and tryptic soy broth (TSB) were purchased from MilliporeSigma (Burlington, MA, USA). Brain heart infusion (BHI) broth was purchased from Research Products International (Mt Prospect, IL, USA). Magic Bullet blenders, cups, and blades were purchased from Amazon.com (Seattle, WA, USA).

A *Staphylococcus epidermidis* Monoclonal Antibody (primary, MA135788) was purchased from ThermoScientific (Waltham, MA, USA). A Goat Anti-Mouse IgG Alexa Fluor 488 Polyclonal antibody was purchased from SouthernBiotech (Birmingham, AL, USA). ProLong Gold Antifade Reagent Liquid was sourced from Cell Signaling Technology (Danvers, MA, USA). The blocking buffer was created using 1% Bovine albumin, 0.1% Tween 20, and 0.1% Triton-X 100 (Millipore Sigma/Sigma-Aldrich, Burlington, MA, USA) diluted in phosphate-buffered saline (PBS). Visualization was conducted using a Leica DMi8 inverted microscope purchased from Leica Microsystems (Wetzlar, Germany). Surgical tools, histological processing equipment/materials (water bath, slides, cover slides, and chemicals), and other miscellaneous supplies were provided by the Bone and Biofilm Research Lab (Salt Lake City, UT, USA).

### 2.2. Skin Sampling Methodology and Sample Collection

Seven female farm pigs (Yorkshire/Landrace hybrid) from Premier BioSource (San Diego County, CA, USA) were euthanized as part of a separate study with approval and oversight of the Institutional Animal Care and Use Committee (IACUC) at the University of Utah (Protocol Number 21-09023, approved 29 September 2021). Necropsy began within approximately 30 min of euthanasia administration. Hair was clipped along the back of each pig. Each pig was marked according to the diagram in Figure 1 with five locations (zones) down the back. Wiping was performed on the skin’s surface at least three times with sterile water and gauze to remove visible debris. One side of the pig received an alternating alcohol-CHG surgical scrub, leaving the opposite side as a baseline control. PSP was applied using the current clinical standard of care and according to the manufacturer’s instructions; the CHG solution was applied on the specified sites in concentric circles alternating with 70% isopropyl alcohol (each 3×) and allowed to dry for 5 min. The cup scrub and tissue blend methods were then performed on each side of the pig.

The cup scrub method was performed using sterile, cylindrical cups (1” inner diameter) placed on the skin. With the cup held securely in place, 5 mL of sterile D/E broth was added to the cylinder with a pipette, 1 mL at a time to minimize splashing. A rubber policeman was used to rub the skin within the cup for approximately 1 min. Strokes across the skin were made randomly. The liquid was mixed, and 1–2 mL of the D/E broth was removed with the pipet and placed in a sterile tube. The tube was capped and put in a chilled cooler. Any remaining broth was discarded, and the cup was removed.

Full-thickness tissue samples were collected as previously described [48]. In short, the skin samples were harvested using the aseptic technique: at least 1 cm apart and at least 5 cm away from the spine (Figure 1). The full-thickness samples were placed individually into conical tubes (50 mL) filled with 35 mL of D/E broth to neutralize the CHG (Figure 1). The samples were placed and transported in a chilled cooler to mitigate bacterial replication.

Half of the samples were treated with a CHG PSP, and the other half were treated with sterile water (no antiseptic) to detect native pig flora (control). In total, 35 CHG-treated cup scrub samples, 35 CHG-treated tissue blend samples, 35 control cup scrub samples, and 35 control tissue blend samples were collected (5 samples per treatment group for each pig × 7 total pigs). The samples were harvested using the aseptic technique. Instruments were sterilized on-site using a glass bead sterilizer between locations within the same test group and between groups.

### 2.3. Sample Processing and Bioburden Quantification

Personal blender cups and blades were cold sterilized using a similar process to previous work [48]. We exposed all internal parts of each blender cup and base to 200-proof ethanol for >6 h. Blenders were run with ethanol for at least 15 s. Between use and before the subsequent sterilization, the blender cups and blades were boiled for approximately 1 min. Before use, the ethanol was discarded, and each blender cup was filled with sterilized deionized water. The water was blended for 15 s to remove/dilute residual ethanol.

Each full-thickness tissue sample and suspending D/E liquid was transferred to a sterile blender cup and blended for approximately 5 min. Blending was performed in intervals to avoid heating the sample solution. Blender cups were manually agitated up and down to ensure thorough homogenization. Tissue blend samples were transferred back into their respective conical tubes. Each tube was vortexed for 1 min and sonicated for 10 min.

Cup scrub samples were also vortexed for 1 min and sonicated for 10 min. Two hundred µL of each sample mixture were plated and spread with a sterile loop on Columbia blood agar to make a 0-dilution plate. Bioburden was quantified using a 10-fold serial dilution series: 100 µL of the sample mixture was serially diluted in 900 µL of PBS carried through to 10^−4^. Aliquots of each dilution (5 × 10 µL) were plated on Columbia blood agar. The sample weight (if applicable), D/E volume, CFU, and surface area were used to quantify the CFU/cm^2^ of each sample (except in the case of pig 1 where the sample weights were estimated). The plates were incubated under aerobic conditions for 48 ± 4 h at 37 °C in a jacketed incubator. Colony counts were used to represent the bioburden in units of CFU/cm^2^.

### 2.4. Process Control Samples

The tissue blend method was also performed without pig skin to verify that each processing step was void of contamination. The blenders were sterilized according to the same procedure outlined above. The D/E broth was blended, transferred to conical tubes, vortexed, and sonicated, as previously described. CFU counts, if any, were determined in the same way as the tissue samples.

### 2.5. Isolate Characterization

Bacterial colonies from each sample were classified according to morphology and color. Representative colonies of each distinct morphology from each of the four study groups (CHG tissue blend, CHG cup scrub, control tissue blend, and control cup scrub) were isolated with sterile loops, streaked onto TSB agar, and incubated for approximately 48 h at 37 °C. Each bacterial isolate was cataloged and cryopreserved (−80 °C).

For further identification and Gram staining, bacterial isolates were cultured on TSB agar and incubated at 37 °C for 24–48 h. Each selected isolate was characterized by Gram staining and light microscopy. Additionally, we used the catalase test to categorize the isolates further. All organisms were cross-compared using Gram staining, organism shape, colony morphology, colony color, catalase classification, and previously identified isolates using genotypic identification [48].

### 2.6. Location Analysis

We divided the pig back into zones from the neck to the rump. These zones were drawn on the pig back (visible in Figure 1, vertical numbering in panel A). We reorganized the bioburden (CFU/cm^2^) from each treatment group into 5 location zones: zone 1 (closest to the neck), zone 2 (below zone 1), zone 3 (middle zone), zone 4 (below zone 3), and zone 5 (closest to the rump). We observed the data trend across zones 1–5.

### 2.7. Histology

Two skin samples from the same pig were fixed in 4% paraformaldehyde for approximately 20–36 h. The first sample was collected from an untreated control area below zone 5. The second skin sample came from a CHG-treated area above zone 1. We dehydrated both samples in increasing ethanol concentrations for a minimum of 2 h in each concentration (70%, 95%, and 100%) and embedded each sample in paraffin wax using a Tissue-Tek 6 AI vacuum infiltration processor (Sakura Finetek USA, Inc., Torrance, CA, USA). The samples were cut to approximately 5 µm sections on a Leica HistoCore Autocut R from Leica Biosystems Nussloch GmbH (Nussloch, Germany). Once sectioned, each sample was mounted to a glass slide. Each slide was washed in 100% xylene for 2 × 10 min to deparaffinize and rehydrated in decreasing concentrations (95%, 70%) of ethanol. We rinsed each sample with deionized water and let it sit in PBS for 10 min. In a similar way, positive and negative control slides were also prepared using *S. epidermidis* ATCC 35989 grown on a collagen plug.

We incubated the slides in a blocking solution consisting of 1% Bovine Serum albumin, 0.1% Tween-20, and 0.1% Triton-X 100 diluted in PBS for 1 h. We then incubated the slides with a 1:100 dilution of an *S. epidermidis* Monoclonal Primary Antibody in PBS at room temperature for 1 h. Previous work investigating microbial species in farm pigs showed *S. epidermidis* to be a primary skin colonizer [48]. The *S. epidermidis* Monoclonal Primary Antibody was chosen based on the relative distribution of *S. epidermidis* on pig skin [48]. Samples were washed in PBS. We performed a secondary incubation with a 1:500 dilution of the Goat anti-Mouse Alexa Fluor Plus 488 Secondary Antibody in PBS at room temperature for 1 h. This incubation was performed in the dark to prevent photobleaching. We washed the incubated samples in PBS 3x and cover-slipped each slide using Prolong Gold Antifade to preserve the immunofluorescent (IF) stain. Slides were visualized using light microscopy We employed a green light filter and overlayed the fluorescent image with real light images.

### 2.8. Statistical Analysis

Multiple locations on each pig’s back were used for each sampling technique, which introduced data clustering. Therefore, we analyzed the data using a mixed effects linear regression (a multilevel model), accounting for locations nested along the pig’s back [48]. In our model, the experimental condition was a fixed effect, and the pig was a random effect. Given the small sample size, we specified the model to use a significance test based on a t statistic, rather than the default z statistic, and fitted using Stata-17 statistical software (StataCorp LLC, College Stata, TX, USA). The following comparisons were performed: control tissue blend vs. CHG tissue blend, control cup scrub vs. CHG cup scrub, control tissue blend vs. control cup scrub, and CHG tissue blend vs. CHG cup scrub.

## 3. Results

### 3.1. Microbial Quantification Outcomes

The tissue blend method captured more bioburden per cm^2^ of porcine skin than the cup scrub method (Figure 2). While CFU counts varied across the data, clear differences were observed between the treatment groups. Both CHG-treated and control samples showed statistically significant differences between the cup scrub and tissue blend methods of sampling. (Figure 2). Process control data indicated that spurious contamination was negligible.

The difference between the cup scrub and tissue blend methods was especially pronounced following CHG PSP from the left side of the pig’s back (see Figure 2, blue bars). Specifically, the cup scrub method yielded 1.05 ± 0.24 log_10_ CFU/cm^2^, while the tissue blend method resulted in 3.24 ± 0.24 log_10_ CFU/cm^2^, more than a 2 log_10_ difference (*p* < 0.001). A bioburden count of 0 signified that the total CFU/cm^2^ was below detectable levels. This phenomenon was observed following the application of a CHG PSP in 5/7 pigs using the cup scrub method (Figure 3). In contrast, a result of 0 (below detectable levels) was observed in only 1/7 pigs following CHG PSP using the tissue blend method (Figure 3). Without exception, the tissue blend method detected more bacteria than the cup scrub method following a PSP in every pig analyzed (Figure 3).

Though not as pronounced, differences between the cup scrub and tissue blend methods were also observed in the control groups. Using the cup scrub method, the average bioburden per cm^2^ of control skin samples was 2.62 ± 0.21 log_10_ CFU (Figure 2, gray bars). Using the tissue blend method, the bioburden of control pig flora was 3.46 ± 0.24 log_10_ CFU/cm^2^ (Figure 2, gray bars). This difference was 0.84 ± 0.45 log_10_ CFU/cm^2^ and this was statistically significant (*p* < 0.001). As with the PSP-prepared side, the tissue blend method detected more bioburden per cm^2^ in every pig participating in this study, although this difference was nearly comparable in pig 4 (Figure 3). Neither method in the control group produced a sample with a bioburden below detectable limits (Figure 3).

Within methodology types, statistical equivalence was observed between the CHG-prepared skin and the control skin processed using the tissue blend method, but not the cup scrub method. Looking across each animal, there was no observable trend between the CHG PSP samples and the control tissue blend samples (Figure 3). The comparison between these groups was not statistically significant (Figure 2; *p* = 0.321). In contrast, the difference in bioburden between the CHG and control cup scrub samples was statistically significant (Figure 2; *p* < 0.001). Across the animals, the difference between these groups ranged from about 1 log_10_ CFU/cm^2^ to more than 2 log_10_ CFU/cm^2^.

### 3.2. Representative Isolate Characterization

More microbial variety was observed in the control samples of both methods than in the CHG PSP-treated samples (Figure 4). A greater quantity of unique isolates was found in native pig skin cleansed with water (controls), with 30.1% and 27.9% of all isolates present following the cup scrub and tissue blend methods, respectively. A total of 24.8% of the isolates came from the CHG PSP-treated samples from the tissue blend method and 17.3% came from the CHG cup scrub group. Gram stain data showed that most species cultured were Gram-positive cocci (82.7%). Some species were Gram-positive bacilli (12.8%). Other microorganisms did not grow after cryopreservation, did not stain, or did not fit into any of the other categories (i.e., coccobacillus). One isolate was identified as Gram-negative (0.4%). Most isolates were catalase-positive (94.7%).

### 3.3. Anatomical Analysis

We observed a mild correlation across the anatomical zones assigned vertically along the pig’s back. While large differences in CFU counts were not observed between each anatomical zone, there was a general “U” shape in the bar graphs (Figure 5). More specifically, there were slightly more CFU counts per cm^2^ of skin in zones 1 and 5 compared to zone 3 (Figure 5). The values in zones 2 and 4 generally fell in between the end zones (zones 1 and 5) and the middle zone (zone 3). This pattern was more obvious in the bioburden levels of the tissue blend samples. The data within each zone mirrored the results shown in Figure 2.

### 3.4. Histological Analysis

Histological analyses corroborated the microbiological outcomes. For example, IF images of CHG-treated and untreated pig skin showed the presence of bacteria (*S. epidermidis*) in superficial and deep skin regions (Figure 6). Bacteria were observable with and without the application of CHG PSP, especially along the hair follicle tract. More bacteria were observed near the surface of the untreated pig skin than on the CHG-treated skin. We observed pockets of bacteria in the dermis of both samples.

Positive and negative controls were also performed (see Appendix A). An IF signal was observed on the positive control sample with *S. epidermidis* ATCC 35989 grown on a collagen plug. No signal was observed on the negative control incubated without the primary antibody.

## 4. Discussion

The purpose of this study was to compare the bacterial bioburden in pig skin between the cup scrub and tissue blend [48] methods following PSP. Our hypothesis that the cup scrub method would not resolve bacteria in all skin layers was supported. Samples quantified using the tissue blend method consistently cultured more bacteria from all skin regions compared to the cup scrub method. IF images corroborated the tissue blend method data, showing that *S. epidermidis* was present in deep skin regions with and without PSP treatment. These outcomes underpinned the potential benefit of using a destructive skin sampling method to test new PSP technologies.

The bacterial sampling methodology directly influenced bioburden levels after PSP application. Using only the cup scrub method applied to pigs, CHG PSP approached the FDA-required 2 log_10_ reduction for PSPs [36] for dry areas with a log_10_ reduction of 1.57 ± 0.45 log_10_ CFU/cm^2^. Compared to the control cup scrub data, this bacterial reduction was statistically significant (*p* < 0.001). In contrast, the CHG PSP kit did not meet the FDA-required reduction when the tissue blend method was used with a log_10_ reduction of 0.22 ± 0.48 log_10_ CFU/cm^2^. Using the tissue blend method, the PSP-treated skin was statistically insignificant compared to the control skin wiped with sterile water (*p* = 0.231). This result suggested that when bacteria from all layers of the skin were considered, CHG as a biocide was statistically equivalent to a sterile water presurgical scrub without chemical disinfectants. While CHG appears to exhibit promising outcomes as a PSP using the cup scrub method, the tissue blend method leaves questions as to its true efficacy below the surface.

While considerable variation was observed between and within animals, overall trends in the data between methods were reflected in the individual results of all seven animals. Across the seven pigs included in this study, the CHG-treated cup scrub samples consistently exhibited the lowest bioburden, even when the CHG-treated tissue blend samples exhibited bioburdens of several log_10_ CFU/cm^2^ higher. Further highlighting the limitations of the cup scrub method, 5/7 pigs had at least one sample where the bioburden was below detectable limits when the cup scrub method was used following CHG application. This was 5× the frequency of the CHG-treated tissue blend group. The bioburden results from each animal individually highlighted the difference between the cup scrub and tissue blend methods.

This discrepancy between the cup scrub and tissue blend methods following PSP was due to the depth of skin each method considered. PSP chemicals were applied to the skin’s surface where microbes were rapidly killed upon application. The cup scrub method principally detected surface-dwelling bacteria impacted by PSP. In contrast, the tissue blend samples captured up to 1–2 cm of skin and subcutaneous fat (full-thickness tissue). Thus, the tissue blend method detected bacteria from all layers of the skin, including pockets found along dermal features such as hair follicles and glands. Additionally, cylinders used in the cup scrub method produced a sampling area of approximately 5 cm^2^, whereas each tissue blend sample encompassed approximately 16 cm^2^. This larger surface area led to the inclusion of a greater number of dermal features, thus resulting in a more accurate average. Across greater depths and larger surface areas, the tissue blend processes accounted for bacteria in more locations than the cup scrub method.

Based on these data, we estimated the effectiveness of the tissue blend method and proposed a relative bacterial distribution. Operating under the assumption that most bacteria on the surface of pig skin were eradicated using CHG PSP, the hundreds to millions of CFUs detected after applying the tissue blend method came from below the surface. These bacteria were not resolved using the cup scrub method. Specifically, there was more than a 2 log_10_ difference between the bioburden of the CHG-prepared cup scrub and tissue blend samples. This result indicated that the tissue blend method applied in domestic pigs may be over 100 times more sensitive to bacteria following the application of CHG PSP than the cup scrub method. The data suggested that more than half of porcine flora live under the surface of the skin. This conclusion in pigs was similar to previous work investigating the scraping and swabbing methods and destructive methods in humans [22].

Histological analysis of porcine tissue supported the microbiological outcomes, including bacterial distribution throughout skin layers. *S. epidermidis* was selected as the representative species as it is a common skin commensal in both pigs and humans [48,49]. This decision was informed using previous genetic sequencing information from samples collected using the tissue blend model in pigs [48]. Generally, the IF signal was localized at the surface of the skin and along the hair follicle (base and shaft). While the signal intensity varied from panel A to panel B, this was likely due to microbiota differences between the sample collection sites more than any impact of PSP application. The sample from panel A was from a posterior region below zone 5, with frequent proximity to fecal matter. The sample from panel B was taken above zone 1 from a CHG-prepared region close to the neck of the pig. In this study, we specifically tagged *S. epidermidis* even though it is just one of many natural colonizers of pig skin. Previously, we found *S. epidermidis* to be a primary colonizer of pig skin [48]. It is likely that there were various other primary colonizers of the area close to the pig neck (panel B) that the *S. epidermidis* IF protocol did not detect. Notwithstanding differences in location, an IF signal from *S. epidermidis* was observed along hair follicle shafts and in the subcutaneous tissue on both samples, suggesting the need for models such as the tissue blend method.

Anatomical location was a major consideration in this study. We designed the collection of each sample type along vertical columns down the pig’s back. When we analyzed the bioburden quantity from zone 1 to zone 5, we detected a higher quantity of bacteria at the peripheries of the dorsum, at the neck and rump, compared to the center of the back, indicating some anatomical variability in pig-back bioburden. Notwithstanding differences across zones, the same trends between the cup scrub method and the tissue blend method were observed.

While data from this study showed that the cup scrub method has limitations, it may be useful in some cases. The cup scrub method was previously used successfully in many investigations spanning clinical dermatology and cosmetic research in addition to antisepsis [39,40,41,42,43,44]. This non-destructive technique is important for researchers in a clinical environment when biopsies and preclinical models are impossible to incorporate; 16 cm^2^ wounds are not appropriate for patient volunteers. Thus, the path of this methodology from research to an ASTM standard to a regulatory document outlining PSP approval is understandable. However, by endorsing the standard and encouraging its use for PSPs, the FDA has incorporated a significant oversight into its approval process. Unfortunately, this has led the scientific community to believe that this standard accurately represents skin flora. Data from this study showed that this is not true; our results showed that a diverse and significant quantity of the natural flora was found out of reach of this skin sampling method. While the cup scrub method may be an acceptable method, its limitations must be recognized when developing new PSP methodologies.

The institutionalization of the cup scrub method may be hampering the development and innovation of PSP products and techniques. We previously called out this issue as a “blind spot” in PSP testing [48]. The lack of consideration given to deep-dwelling bacteria is concerning because the bacteria that evade PSP application are the colonizers of surgical wounds and thus the culprits of SSIs. We conclude that CHG as an antiseptic does not diffuse adequately within the time allotted in the operating room (<5 min) and may need future consideration. The early phase development of the next generation of PSPs might be aided by pivoting away from clinical testing towards a variation of our porcine model using full-thickness skin sampling.

This study was not without limitations. By necessity, we collected samples after euthanasia. Cup scrub and tissue blend sample processing was initiated as soon as all specimens were collected and transported to the lab for analysis (a 7 min walk). We minimized bacterial replication by beginning analysis immediately after sample collection and storing the samples in a chilled cooler to prevent log phase growth. If bacterial replication had occurred, it likely would have been reflected equally across all samples and would not affect the overall study outcomes. Additionally, the representative isolates collected from the samples were not meant to be comprehensive, only representative of the types of microorganisms seen. The genetic identification of pig skin species was performed previously and not repeated for this investigation [48].

We acknowledge that this protocol was created in alignment with ASTM standard E1173-15 but was modified in specific ways due to the nature of our animal investigation [36]. First, ASTM E1173-15 is a human standard that we adapted for a porcine model. This standard specifies that threshold testing should be performed before the cup scrub method is used. We did not perform this screening as we could not exclude any of our animals based on the initial bioburden. Additionally, a D/E neutralizing broth was used as the suspension solution to neutralize the CHG upon skin contact instead of the solution proposed in the ASTM standard to avoid false negative cultures. Finally, the standard recommends clipping 48 h before testing. Clipping was performed immediately before the collection process to avoid unnecessary animal anesthesia. We removed all hair and debris during the water step. This choice was made specifically to mirror how human skin is prepared before surgery. We are currently performing a clinical study using discarded skin from reconstruction surgery to test the clinical translatability of the data collected in pigs.

The tissue blend method applied in pigs was useful in detecting bacteria in all skin regions following PSP. We showed that discrepancies in assumed PSP effectiveness were due to variations in sampling techniques. The data uncovered limitations of the ASTM standard E1173-15 for testing PSP products and emphasized the need for and benefit of an animal model for future PSP development. The results of this study were consistent with our previous work on the tissue blend method [48]; previous animal investigations and the data from this study exhibited considerable overlap. Additionally, IF imaging supported our microbial quantification data. As the tissue blend method applied in a porcine model continues to be used and developed, we encourage researchers and PSP manufacturers to incorporate full-thickness sampling as part of their testing strategy. The tissue blend method is a robust methodology that may be used to create next-generation PSP technologies.

## 5. Conclusions

This study suggested that the cup scrub method, a clinical standard for antiseptic approval based on ASTM standard E1173-15, may be underreporting the quantity of bacteria surviving PSP, leading surgical staff to believe PSP-treated skin is “sterile”, when, in fact, it is not. When developing new PSP technologies, the tissue blend method applied in pigs may better resolve true biocidal efficacy in deeper skin regions.

## Figures and Tables

**Figure 1 microorganisms-12-02369-f001:**
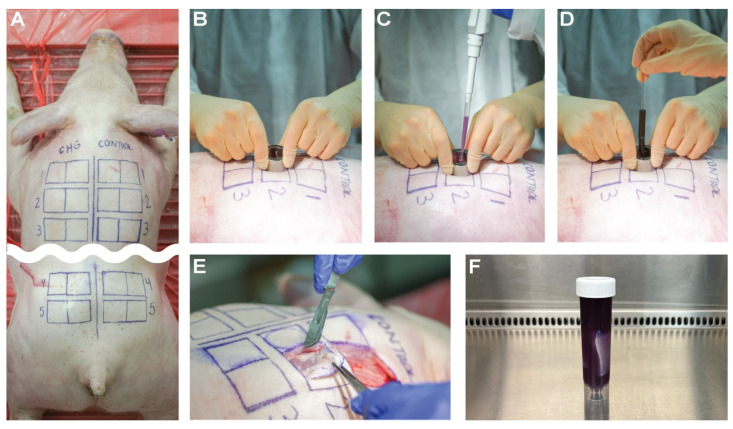
Images of the sample collection methodology. (**A**) The pig’s back was bisected along the sagittal axis. The left side of the back was treated with a CHG PSP. The right side of the back was cleansed with sterile water to serve as the control (native flora). Boxes (4 × 4 cm) were drawn with a sterile skin marker. Medial boxes defined the location for the cup scrub method and lateral boxes defined the area to be excised for the tissue blend method. (**B**) Pictured is a cylinder employed in the cup scrub method and secured to the skin with pressure. (**C**) D/E broth (5 mL) was pipetted into the cup. (**D**) The skin was agitated with a rubber policeman for 60 s. Endogenous flora was dislodged and suspended in the D/E broth during agitation. Two mL of the D/E broth were removed via a pipette for quantification. (**E**) Tissue in the lateral boxes was excised using a sterile scalpel and forceps. Excised tissue was transferred to a sterile 50 mL conical tube. (**F**) Excised skin in D/E neutralizing broth ready for homogenization via blending.

**Figure 2 microorganisms-12-02369-f002:**
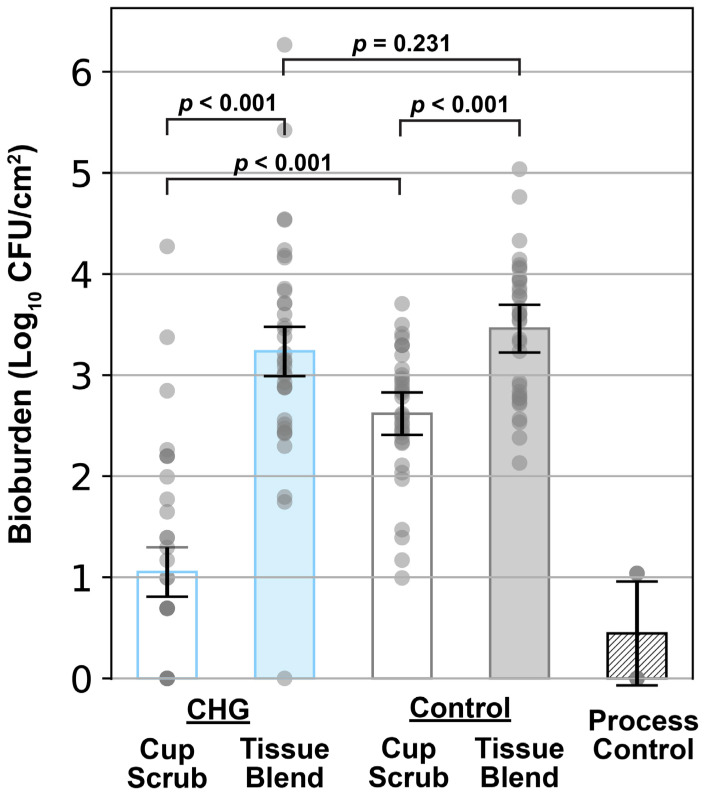
The bioburden (Log_10_ CFU/cm^2^) determined in 7 pigs and differentiated by treatment group and processing method. The blue bars represent CHG treatment (the left side of the pig’s back). The gray bars represent the control samples (no antiseptic; right side of the pig’s back). The process control bar represents all steps of the tissue blend method without skin present. The gray circles represent individual data points from each sample. The error bars show the standard error for the indicated treatment group (n = 5 per pig).

**Figure 3 microorganisms-12-02369-f003:**
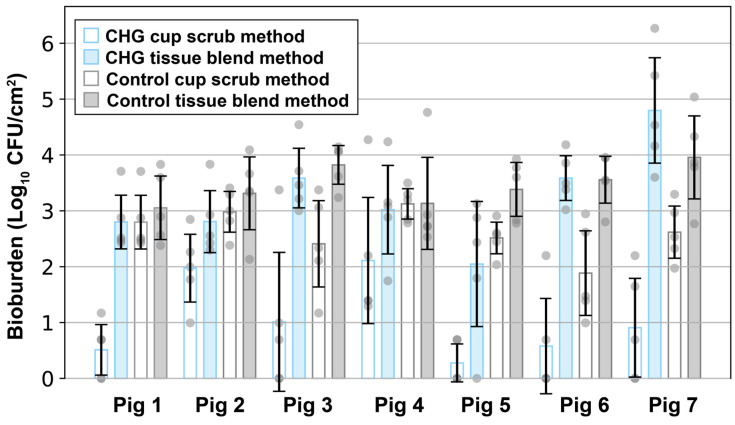
The outcomes of each individual pig indicating bioburden (Log_10_ CFU/cm^2^) were plotted as a function of treatment group, sampling methodology, and animal. The error bars represent the standard deviation of the data set per treatment. Each gray circle indicates a single sample.

**Figure 4 microorganisms-12-02369-f004:**
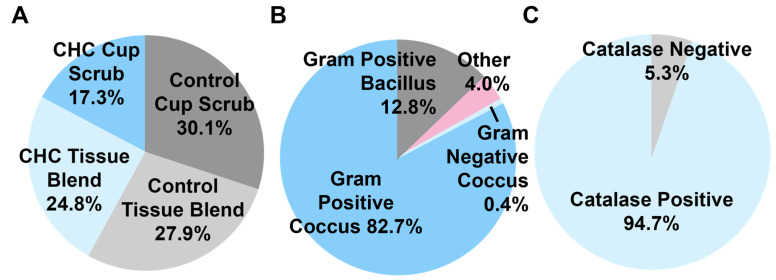
A breakdown of representative bacterial isolates cultured from each of the various treatment groups (*n* = 226). (**A**) The quantity of isolates cultured from each group (%). (**B**) A breakdown of the isolates identified by Gram staining. (**C**) The distribution of isolates that were catalase positive or negative. Data rounded to the nearest tenth.

**Figure 5 microorganisms-12-02369-f005:**
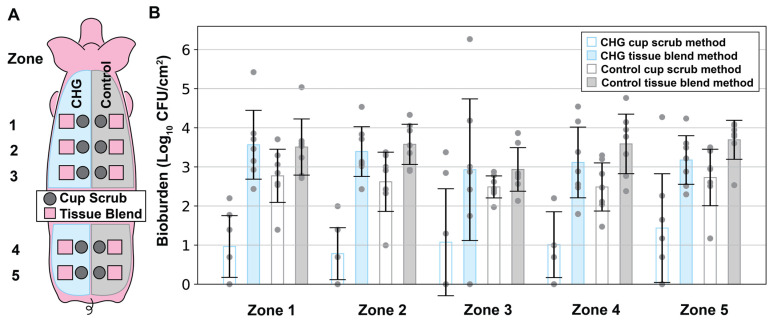
(**A**) The zones along the pig’s back were defined from superior to inferior in numerical order 1–5. (**B**) Bioburden was categorized by the treatment and sampling methods as a function of the anatomical location (zone).

**Figure 6 microorganisms-12-02369-f006:**
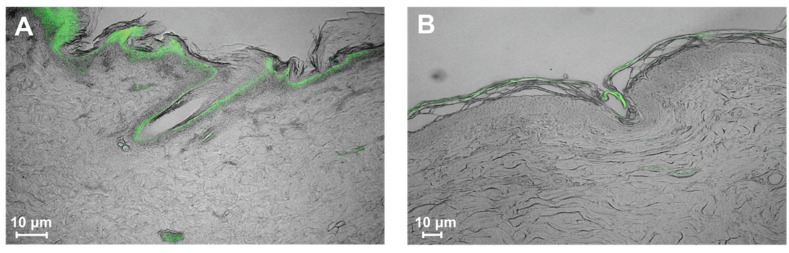
Cross-sections of porcine skin stained with IF for *S. epidermidis*. Fluorescent green indicates the presence of bacteria. (**A**) The control, unwiped pig skin section collected from below zone 5 showed bacterial presence throughout the epidermal and dermal features. (**B**) CHG-treated pig skin collected from above zone 1.

## Data Availability

The original contributions presented in the study are included in the article, further inquiries can be directed to the corresponding authors.

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
