# Peer review of "Regulatory Standard for Determining Preoperative Skin Preparation Efficacy Underreports True Dermal Bioburden in a Porcine Model"

_microorganisms, 2024, doi:10.3390/microorganisms12112369_

Round 1

Reviewer 1 Report

Comments and Suggestions for Authors

The article is consistent and the topic is absolutely relevant. The text is clear and methods well described. 

Discussion is excellent. I would still suggest specifying the post-mortem interval from the death of the pig to the analysis and, possibly, speculating on consequences with respect to the live microbiome.

The only minor issue concerns the references from 38 to 62 line 93. Perhaps it would be more appropriate to cite the main studies or review, but I defer to your judgement as to the appropriateness of defining a sort of state-of-the-art method.

Many thanks

Reviewer 2 Report

Comments and Suggestions for Authors

1.  The authors did not conduct an in-depth analysis of the microbial species detected in each sampling method. Such characterization, especially identification of pathogenic species relevant to surgical site infections, would enhance the clinical significance of the findings.

2. The cup scrub method samples a much smaller area compared to the tissue blend method (5 cm² vs. 16 cm²), which may introduce bias and potentially exaggerate the differences observed. Equalizing sampling areas would provide a more controlled comparison.

3. The background related to skincare could be enhanced by referencing the articles 10.1021/acsmaterialslett.4c00392 and 10.1021/acsami.1c25014. Additionally, a discussion of the advantages offered by the current study over similar works in related fields would be beneficial.

4. The study focuses solely on chlorhexidine gluconate as the antiseptic agent. Testing a variety of PSP agents would provide a broader understanding of the limitations of current ASTM standards and how different antiseptics may penetrate the skin layers.

5.  The histological images focus mainly on S. epidermidis, which is only one of many bacterial species found on the skin. Expanding the analysis to detect other bacterial species in situ would give a fuller picture of bioburden in deep skin layers.
